# Self-reported musculoskeletal disorders questionnaire for agriculturists: An online self-assessment tool development

Worawan Poochada[1], Sunisa Chailklieng[2,3]*

1 Epidemiology and Biostatistics, Faculty of Public Health, Khon Kaen University, Khon Kaen, Thailand, 2 Department of Environmental Health, Occupational Health and Safety, Faculty of Public Health, Khon Kaen University, Khon Kaen, Thailand, 3 Research Center in Back, Neck, and Other Joint Pain and Human Performance (BNOJPH), Khon Kaen University, Khon Kaen, Thailand

* csunis@kku.ac.th

**Data Availability Statement:** All relevant data are within the manuscript and its Supporting Information files.

## Abstract

The musculoskeletal disorders (MSDs) severity and frequency questionnaire (MSFQ) has been used with agriculturists. Although it frequently appears in the literature, it does not periodically reflect the levels of MSDs. This study aimed to develop a MSDs assessment tool for agriculturists via an online database. The four hospitals that were assigned to the development and tryout group received a random selection of participants from 33 health promoting hospitals. In the development phase, 55 agriculturists (from two hospitals) completed the structured interview questionnaire. The MSFQ document for the analysis of MSDs symptoms among agriculturists was concentrated on the frequency and severity of symptoms. The online MSFQ was checked by using intraclass correlation coefficients (ICC) in a one-way random-effects model. In the tryout phase, a group of 44 agriculturists (from another two hospitals) completed a similar online MSFQ. Cronbach's alpha was used to test the reliability of the online self-reported MSDs questionnaires. A Likert rating scale, used to measure the satisfaction of users, was divided into three categories: information, program design, and benefits. The development phase showed an ICC = 0.99 with a 95% CI = 0.98–0.99. More than 65 percent of agriculturists were female, aged from 41–50 years old. A large number of them were engaged in self-cultivation. Their main crops were rice and cassava. In total, 50% reported that they had experienced mild MSDs levels. The shoulder, knee-calf, and hip-thigh areas were the top three parts of the body where agriculturists had experienced MSDs from cultivation. Excellent reliability of online MSFQ was found after the tryout. The agriculturists were very satisfied overall (information, program design, and benefits). Developments to the MSFQ and online administration did not diminish its reliability for obtaining information about the level of MSDs of agriculturists' musculoskeletal symptoms. This self-reported MSDs questionnaire was appropriate to be used among agriculturists.

**Funding:** This study received funding from The National Research Council of Thailand (NRCT 6200101) to SC.

## Introduction

Thai agriculturists have been found to make up the majority of informal workers and almost all of them have been engaged in plant cultivation [1]. Musculoskeletal disorders (MSDs), which had been reported as occupational diseases in the Health Data Center (HDC) database of Thailand, had the highest ranking of all diseases among agriculturists who visited healthcare service providers [2]. The prevalence of MSDs among agricultural workers was as follows: 87.9% of people reported experience of MSDs in the previous 7 days [3], 70.30% of people reported experience of MSDs in the previous month [4], and 88.9% of people reported experience of MSDs in the previous 12 months [3]. A systematic review showed MSDs had the highest prevalence rates of work-related diseases (67.80%, 95%CI 66.30–69.30) among agriculturists in Thailand [5]. The duration of disability caused by MSDs ranged from 1 to 190 days [6]. Lower extremities, lower back, and shoulders were the three body parts with the highest prevalence rates of MSDs [3, 4]. Cultivating agriculturists working on cassava, fruit, vegetable, and corn plantations had a significantly higher risk of knee/calf pain (OR = 1.97, 95% CI = 1.35–2.89) and lower limb pain (OR = 1.97, 95% CI = 1.37–2.84) than those working on rice and tobacco plantations [4]. Working conditions such as using agricultural tools, prolonged static postures, and lifting >10 kg have been found to be related to agriculturists' health [3]. MSDs are the key priority issue for occupational and environmental health problem surveillance among agriculturists.

The assessment of exposure to risk factors for work-related MSDs can be performed by using three methods: 1) observational methods, 2) self-reports, and 3) direct measurement by specific measurement, i.e., using a goniometer and electromyography (EMG) [7]. Self-reported assessment has quite commonly been used to measure symptoms, including pain and postural discomfort, and/or levels of subjective exertion. The Nordic Musculoskeletal Questionnaire (NMQ) by Kuorinka et al. [8], which has mostly been used in epidemiological research and several occupations, asks the participants to report the bodily complaints that they have experienced in the past 7 days and the past 12 months. However, the NMQ does not assess the severity or frequency of the bodily discomfort or pain. In Thailand, a self-reported MSDs questionnaire named "the MSDs Severity and Frequency Questionnaire" (MSFQ) was developed by Chaiklieng [9] for use in Thai industrial labor, i.e., potato-chip workers [9], electronics workers [10], pulp and paper production workers [11], and rolled steel roofers [12]. The MSFQ has been used with the interview method in the agricultural sector with workers engaging in various cultivation activities (rice, tobacco, sugarcane, cassava, fruit, vegetable, corn, and rubber [4, 13]). The MSFQ contains four main questions about pain which had occurred in bodily areas during work or had been caused by work in the past month: (1) severity of pain, (2) frequency of pain, (3) work-related pain in the last 7 days, and (4) confirmation of the work-related pain. The MSFQ's five MSDs levels were determined by multiplying the severity level by the frequency of pain level. However, the MSFQ had limitations in regard to immediately reporting the MSDs level to workers.

At present, the self-reported survey study is generally used to collect primary data, categorized into manual (paper-based), and electronic (online-based) data [14]. The National Clinical Research Center (nCRC) provides an online research tool for researchers. It is a modern online survey tools creator which includes questionnaire design, distribution, analysis, and reporting. It can also be used to manage any type of research, ranging from a very simple type, such as a quick survey, to very complex multinational random sampling. The advantages of online surveys include increased geographical spread of respondents, faster responses can be started immediately [14], low cost, and paper reduction. Therefore, this study aimed to develop the MSFQ for agriculturists via an online database.

## Materials and methods

### Study design

This study utilized a research and development design of an MSDs assessment tool for agriculturists via an online database (nCRC). Thailand's Khon Kaen University Ethics Committee in Human Research examined and approved the study procedure (HE632162). All participants gave written informed consent before being enrolled in the study.

### Population and sample

The population consisted of cultivating agriculturists who had access to 33 health-promoting hospitals in the upper northeastern Thai provinces of Khon Kaen, Roi Et, Udon Thani, and Nong Bua Lamphu. The stratified sampling technique was used in the following three steps: First, a health-promoting hospital was randomly selected. Then, the selected health-promoting hospital was assigned to either the development or tryout group. Finally, the participants were chosen by simple random sampling of workers who met the inclusion criteria and had given their consent to their participation in the study.

   The inclusion criteria were being a cultivating agriculturist aged 18 years or older, having received service from a health-promoting hospital, and having consented to being part of the sample. Cultivating agriculturists who had a historical diagnosis of MSDs, or a history of past surgery were excluded. There were 99 agriculturists chosen from four hospitals, which consisted of 55 people in the development phase (two hospitals) and 44 people in the tryout phase (two hospitals) who met the inclusion criteria for participation in this study.

### Research tools

A structured questionnaire was used in this study. The first part asked about the characteristics and health status of the agriculturist. The second part was the self-reported MSDs questionnaire, which the MSFQ was applied from Chaiklieng [9]. The MSFQ consists of four main questions: (1) severity of pain, (2) frequency of pain, (3) work-related pain in the last 7 days, and (4) confirmation of the work-related pain in the past month. The MSDs level was multiplied by four levels of frequency and four levels of the severity of pain. The MSDs levels by MSFQ was classified into five perception levels: no MSDs (score: 0 points), mild MSDs (score: 1–2 points), moderate MSDs (score: 3–4 points), severe MSDs (score: 5–8 points), and very severe MSDs (score: 9–16 points). The MSFQ measurement process is shown in Fig 1. A structured online MSFQ was developed by transforming the MSFQ document via the nCRC tool [9]. It was designed to be user-friendly and allow users to report the MSDs level automatically and quickly.

   After the trial with participants, the user satisfaction with the online MSFQ was estimated. Three subcategories of user satisfaction were identified: benefits, program design, and information.

### Methods

This study was divided into two phases: the development and tryout phase, as shown in Fig 2. In the development phase, participants were interviewed by researcher. All variables from the four main parts of the MSFQ document were used to generate, calculate, and report the MSDs levels via the nCRC tool. The interview data was input into the MSFQ online and the agreement between the document and online MSFQ was measured. The tryout phase was carried out when all variables of the document and online MSFQ were in overall agreement. The online MSFQ was tested by each agriculturist through their smartphone. Finally, user

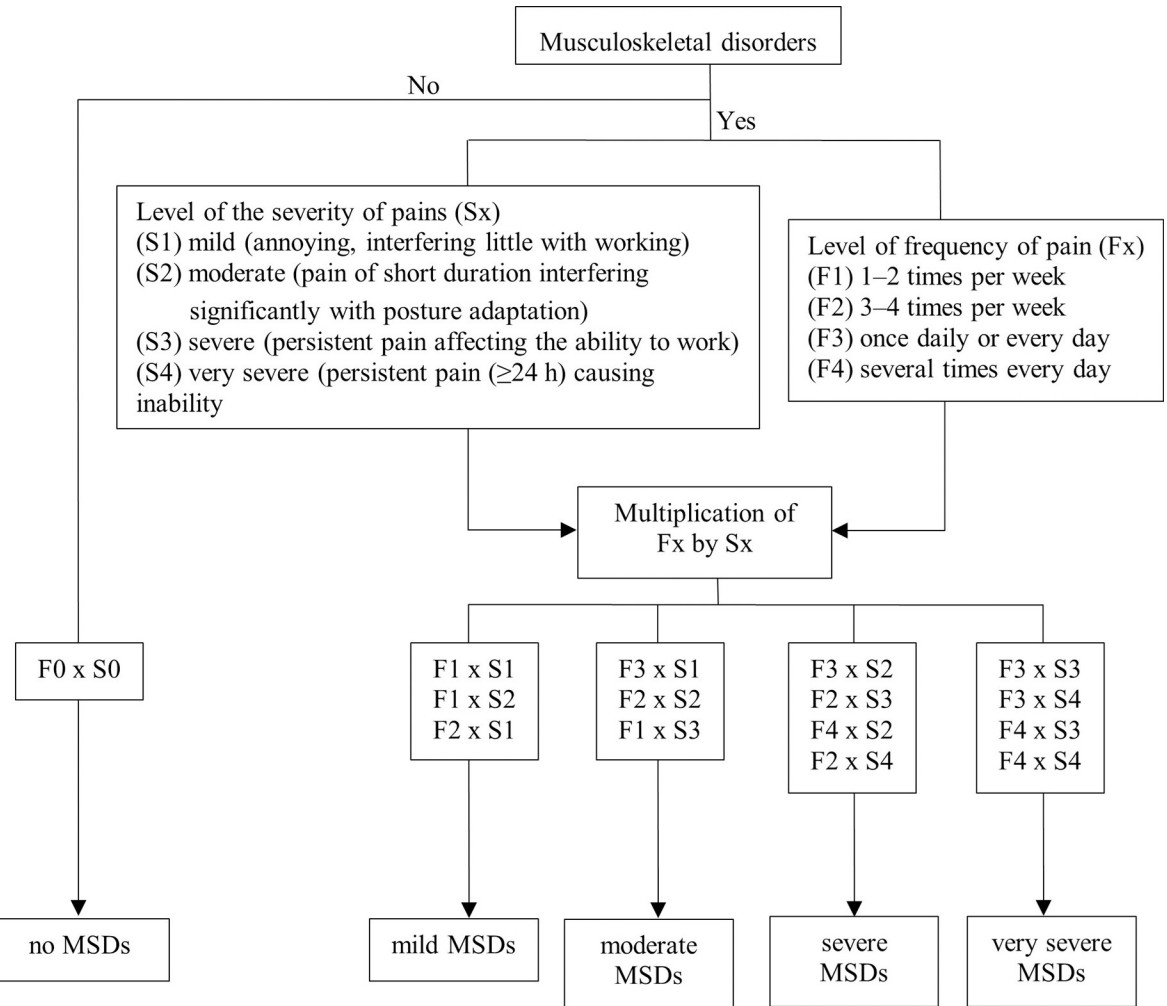

**Fig 1. The MSFQ measurement process (applied from Chaiklieng [9]).**

satisfaction with the online MSFQ created via the nCRC tool was estimated along with performance of the reliability test.

## Statistical analysis

All data analyses were performed using STATA Version 14.0. Descriptive statistics were used to summarize personal characteristics, health status, and MSDs levels. Categorical variables were presented using frequency distribution and percentages and continuous variables were presented using mean (SD) and median (min-max).

The final score of MSDs, or continuous data, was used to analyze agreement by using intraclass correlation coefficients (ICC). ICC values less than 0.50 are indicative of poor agreement, values between 0.50 and 0.75 indicate moderate agreement, values between 0.75 and 0.90 indicate good agreement, and values greater than 0.90 indicate excellent agreement [15].

Cronbach's alpha ($\alpha$) was used to test the reliability of the online MSFQ after the tryout. George and Mallery [16], who are often cited, provide the following rules of thumb: $\alpha > 0.90$ (Excellent), $> 0.80$ (Good), $> 0.70$ (Acceptable), $> 0.60$ (Questionable), $> 0.50$ (Poor), and $< 0.50$ (Unacceptable).

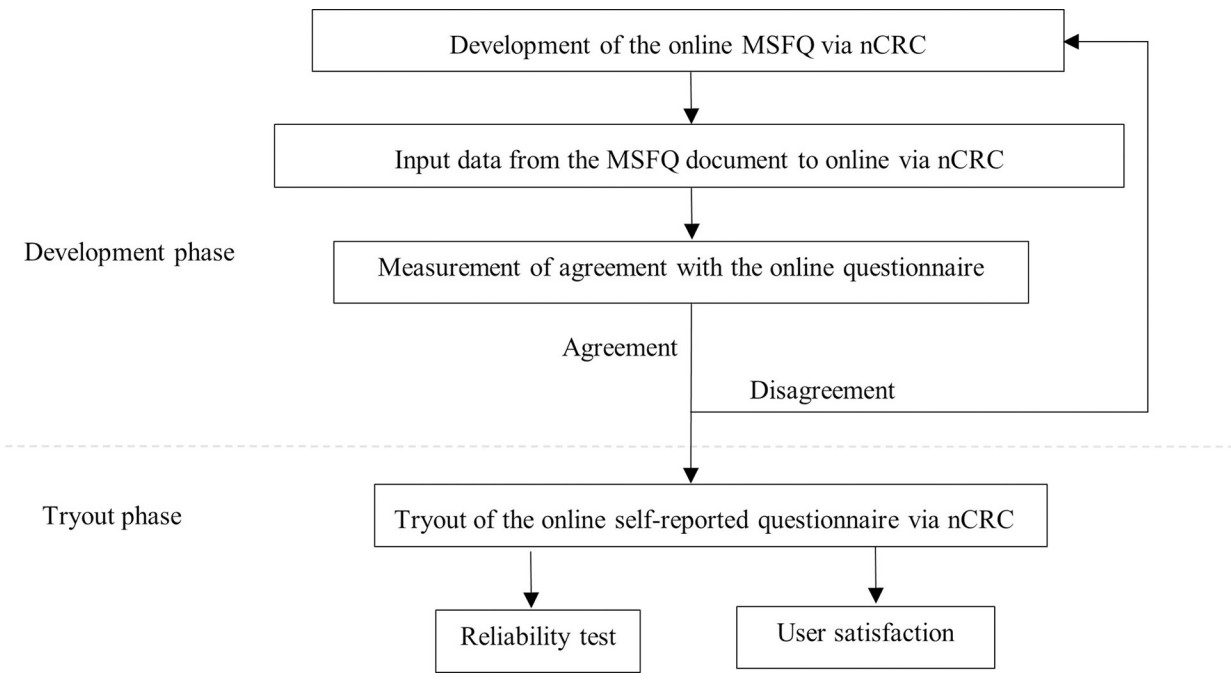

**Fig 2. Flow of the research methodology.**

A Likert rating scale (1–5 points) was used to measure user satisfaction with an online MSFQ via nCRC: very satisfied (5 points), satisfied (4 points), neutral (3 points), unsatisfied (2 points), and very unsatisfied (1 point). The ranges of the average satisfaction score were determined as follows: 0.01–1.00 was very unsatisfied, 1.01–2.00 was unsatisfied, 2.01–3.00 was neutral, 3.01–4.00 was satisfied, and 4.01–5.00 was very satisfied.

## Results

### Development phase

A total of 55 agriculturists from two hospitals were included in the development phase. Most of them (80.00%) were female. Almost 70 percent were aged between 41 and 50 years old. They mostly worked as cultivating agriculturists (89.09%) who cultivated the following crops: cassava (45.45%), rice (40.00%), and sugarcane (14.55%), respectively.

From the document MSFQ, it was found that almost all of the agriculturists (78.18%) reported that they had experienced MSDs in at least one part of the body. The parts of the body where agriculturists had most experienced MSDs were the lower back (30.91%) and knee and calf (30.91%), followed by the lower shoulder (20.00%) and lower arm (18.18%), respectively. Regardless of the part of body, the most experienced level of severity was found to be mild (41.82%) followed by moderate (25.45%) and no MSDs (21.82%), respectively.

The final scores of online MSFQ had excellent agreement. A one-way random-effects model showed an ICC = 0.98 with a 95% confident interval = 0.98–0.94.

### Tryout phase

A total of 44 agriculturists from another two hospitals were included into the tryout phase. Of all the agriculturists, 68.18% were female. Most of them were found to be in the following age ranges: 41–50 years old (40.91%), followed by 51–60 years old (36.36%). A large proportion of them (70.45%) engaged in self-cultivation. They planted rice (70.45%) and cassava (29.55%).

**Table 1. Number (%) of users based on level of satisfaction with the online MSFQ.**

| Topic of satisfaction | Level of satisfaction with the self-reported MSDs questionnaire | | | | | |
|---|---|---|---|---|---|---|
| | VS | S | N | US | VUS | Mean ± S.D. |
| **1. Information** | | | | | | |
| 1.1. Easily understood language for usage | 22 (68.8) | 8 (25.0) | 2 (6.3) | 0 (0.00) | 0 (0.00) | 4.63 ± 0.61 |
| 1.2. The questions are continuous | 23 (71.9) | 9 (28.1) | 0 (0.00) | 0 (0.00) | 0 (0.00) | 4.72 ± 0.46 |
| | | | | | Average | 4.67 ± 0.54 |
| **2. Program design** | | | | | | |
| 2.1. Easy access to appraisals via mobile | 20 (62.5) | 10 (31.3) | 2 (6.3) | 0 (0.00) | 0 (0.00) | 4.56 ± 0.62 |
| 2.2. The font size is suitable | 21 (65.6) | 8 (25.0) | 3 (9.4) | 0 (0.00) | 0 (0.00) | 4.56 ± 0.67 |
| 2.3. Easy to use | 21 (65.6) | 9 (28.1) | 2 (6.3) | 0 (0.00) | 0 (0.00) | 4.59 ± 0.61 |
| 2.4. It has a sequence of steps | 23 (71.9) | 9 (28.1) | 0 (0.00) | 0 (0.00) | 0 (0.00) | 4.72 ± 0.46 |
| | | | | | Average | 4.61 ± 0.59 |
| **3. Benefits** | | | | | | |
| 3.1. Able to self-report MSDs | 24 (75.0) | 8 (25.0) | 0 (0.00) | 0 (0.00) | 0 (0.00) | 4.75 ± 0.44 |
| 3.2. Able to know the MSDs results immediately | 25 (78.1) | 7 (21.9) | 0 (0.00) | 0 (0.00) | 0 (0.00) | 4.78 ± 0.42 |
| 3.3. Stretching muscles was advised | 26 (81.3) | 6 (18.8) | 0 (0.00) | 0 (0.00) | 0 (0.00) | 4.81 ± 0.40 |
| | | | | | Average | 4.73 ± 0.46 |
| **4. Overall** | 20 (62.5) | 11 (34.4) | 1 (3.1) | 0 (0.00) | 0 (0.00) | 4.59 ± 0.56 |

VS = Very satisfied, S = Satisfied, N = Neutral, US = Unsatisfied, VUS = Very unsatisfied

One half of them self-reported that they had experienced MSDs in at least one part of body at a mild MSDs level, while 38.64% had experienced no MSDs. The shoulder, knee-calf, and hip-thigh were the top three parts of the body where agriculturists had experienced MSDs in cultivation; the proportions of agriculturists who had experienced MSDs in these areas were 27.27%, 22.73%, and 20.45%, respectively.

The online MSFQ had excellent reliability, with Cronbach's alpha coefficient at 0.97. User satisfaction with the online MSFQ via an online database was rated at 72.73% (n = 32). Agriculturists were very satisfied overall with the online MSFQ according to the following categories: information, program design, and benefits (see more detail in Table 1). User feedback about the online MSFQ via electronic devices (smartphones) included the following comments: 1) it was difficult to get access via QR code or long URL, and 2) an internet network was necessary to complete this questionnaire.

## Discussions

### Characteristics and MSDs

This study found a higher percentage of women than men in the agricultural sector. It was possible that male agricultural workers were working on farms while we were collecting data. In both the development phase and tryout phase, the highest proportion of agriculturists were aged 41–50 years. Teenage agriculturists may have been hired in the manufacturing sector in urban areas. The crop types of the two phases of the study were similar: cassava and rice. The MSDs reported by MSFQ were in the areas of the lower back, shoulder, knee-calf, hip-thigh, and arm. A previous study found that bodily pain was generally found in the lower back, shoulder, and lower limb areas among cultivating agriculturists in northeastern Thailand [4]. Moreover, MSDs of the knees among Thai agriculturists significantly differed according to farm type (rice, flower, and vegetable) [17].

### Development phase

The MSFQ applied from Chaiklieng [9] was verified. The intra-rater reliability, based on the ICC of the final scores of an online MSFQ, was excellent, with an ICC = 0.98 with a 95% confident interval = 0.98–0.99. Meanwhile, an online MSFQ developed platform was 100 percent accurate with regard to content and MSDs levels calculation when compared to the document MSFQ. The researchers rechecked and improved the online MSFQ to be perfectly accurate because they wanted to develop valid questionnaires for the online platform.

### Tryout phase

This study was conducted to assess the reliability of the online MSFQ developed for evaluating various MSDs at agricultural sites through online MSFQ. This online MSFQ showed excellent reliability, with Cronbach's alpha coefficient as 0.97. Although the participants in this study cultivated cassava, rice, and sugarcane, this online MSFQ can be implemented with those who cultivate other crops. More than 90.00% of Thai agriculturists have completed their primary school education [18], hence communication was understood by them. Agriculturists were very satisfied overall with the online self-reported MSDs questionnaire. The fact that they were very satisfied (4.67 ± 0.54) with the information provided correlated with a previous study; information should be accurate and improvements should always be up to date [19]. Also, they were very satisfied (4.61 ± 0.59) with the program design; a previous study showed that interesting designs used appropriate colors and text sizes, were uncomplicated, and could be followed step by step [19]. Agriculturists were very satisfied with the benefits of the online MSFQ (4.73 ± 0.46). The researchers considered the benefits to agriculturists as a priority.

The MSFQ is only one of three methods used to assess work-related MSDs. It has specifically been used to report pain or discomfort when personal pain descriptions have differed [20]. Loss of muscle strength has been found to affect pain perception. A previous study of rubber planters found that low handgrip strength was a factor which significantly correlated with a high level of MSDs [21]. This study filled the gap by following the study of self-reporting of MSDs by Chaiklieng [9], which expanded on the meaning of severity level, and included the immediate reporting of individual MSDs levels. However, a combination of work-related MSDs exposure assessments was found to be interesting with regard to surveillance of MSDs. Previous studies and self-reported and postural risk assessment using observational methods were combined [21]. Therefore, observational methods should be considered in parallel in terms of the health risk matrix.

## Conclusions and suggestions

This study aimed to develop the MSFQ for agriculturists via an online database. It was reported that 78.18%, and 61.36% of agriculturists had experienced MSDs in at least one part of the body, according to the document and online MSFQ, respectively. It was found that the most experienced MSDs level was mild, followed by moderate, and no MSDs, respectively. The shoulder, knee-calf, and hip-thigh were the top three parts of the body where agriculturists had experienced MSDs in cultivation. The developed online MSFQ was in 100 percent agreement with the document MSFQ. Excellent reliability of online MSFQ was found after the tryout phase. Agriculturists who had estimated their user satisfaction with the online MSFQ reported that they were very satisfied overall (with regard to information, program design, and benefits).

The online MSFQ was appropriate for agricultural users to assess MSDs resulting from their work. In addition, the collection of information in a cross-sectional study in which the researcher will contact the participant only once, and longitudinal studies in which the

researcher should contact the participant more than once, was suitable. The individual MSDs levels found by online MSFQ were immediately reported. However, the online MSFQ had some limitations: the online MSFQ was based on using self-reporting to confirm that the MSDs had been caused by work; this was based on the agriculturist's opinion and there was no diagnosis by a physician. Observational techniques and direct methods are necessary to estimate the root causes that led to the MSDs, such as awkward posture, forceful exertion, repetitive motion, vibration, and work environmental factors. In the future, a mobile application will be developed to solve the problem of agriculturists being unable to access the questionnaire via the internet network. A cohort study should be provided for MSDs surveillance among cultivating agriculturists, along with design of an online-based dashboard to immediately report incidence of MSDs.

## Supporting information

**S1 Fig. The MSFQ measurement process (applied from Chaiklieng [9]).**
(JPG)

**S2 Fig. Flow of the research methodology.**
(JPG)

**S1 Table. Number (%) of users based on level of satisfaction with the online MSFQ.**
(DOCX)

**S1 Appendix An online MSFQ.**
(PDF)

## Acknowledgments

The authors would like to thank all of the cultivating agriculturists and health care officers for their kind co-operation throughout this study.

## Author Contributions

**Conceptualization:** Sunisa Chailklieng.

**Data curation:** Worawan Poochada, Sunisa Chailklieng.

**Formal analysis:** Worawan Poochada.

**Funding acquisition:** Sunisa Chailklieng.

**Investigation:** Worawan Poochada, Sunisa Chailklieng.

**Methodology:** Sunisa Chailklieng.

**Supervision:** Sunisa Chailklieng.

**Validation:** Sunisa Chailklieng.

**Writing – original draft:** Worawan Poochada, Sunisa Chailklieng.

**Writing – review & editing:** Sunisa Chailklieng.

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
