## [Decision Letter · Decision Letter 0]

22 Jul 2022

PONE-D-21-40986Self-reported musculoskeletal disorders questionnaire for agriculturists: An online self-assessment tool developmentPLOS ONE

Dear Dr. Chaiklieng,

Thank you for submitting your manuscript to PLOS ONE. After careful consideration, we feel that it has merit but does not fully meet PLOS ONE’s publication criteria as it currently stands. Therefore, we invite you to submit a revised version of the manuscript that addresses the points raised during the review process. 

The manuscript has been evaluated by two reviewers, and their comments are available below.

The reviewers have raised a number of concerns that need attention. They request additional information on methodological aspects of the study and adjustments to the reporting of the study.  

Could you please revise the manuscript to carefully address the concerns raised?

We look forward to receiving your revised manuscript.

Kind regards,

Johannes Stortz

Staff Editor

PLOS ONE

Journal Requirements:

Reviewers' comments:

Reviewer's Responses to Questions

**Comments to the Author**

1. Is the manuscript technically sound, and do the data support the conclusions?

Reviewer #1: Yes

Reviewer #2: Yes

2. Has the statistical analysis been performed appropriately and rigorously? 

Reviewer #1: Yes

Reviewer #2: Yes

3. Have the authors made all data underlying the findings in their manuscript fully available?

Reviewer #1: Yes

Reviewer #2: Yes

4. Is the manuscript presented in an intelligible fashion and written in standard English?

Reviewer #1: No

Reviewer #2: Yes

5. Review Comments to the Author

Reviewer #1: Comments to the Author:

-The manuscript is very good and valuable and the authors have done a lot of effort, but for me, the question is whether self-report and questionnaire without clinical examination accurately determine that the disease is caused by work?

-the following are suggested for improvement:

• Overall:

-There are some grammatical issues throughout the paper that would benefit from an additional review and in other words grammar needs to be corrected throughout the manuscript.

- Introduction needs to be rewritten.

-The Materials and methods is long and vague, it is better to be concise and clear.

• Abstract:

-The research method (Statistical population, Sampling method, etc.) should be written.

-What tests have been used?

• Instrodation:

-The introduction and problem statement is very brief and requires further explanation.

- The importance of the topic should be explained in such a way that the designed tool can determine whether the disease is caused by the work or not.

- It would be better to include a literature review.

- Research innovation to be written

• Resarch metod:

-The research method should be expressed in a more coherent and clear way

• Discussion

-the discussion should be written to reflect the results

• Conclusion

- The conclusion is very brief so it should be reinforced

- Research limitations, the strengths and weaknesses of the study, suggestions should be written

• References

-The reference should be given at the end of the manuscript according to the journal format.

- Match the references inside the text and at the end of the text.

Reviewer #2: The tool will help in self reporting of MSDs in agriculture and will help to cover workers all around the globe. It is always not possible to find musculoskeletal problems of farm workers experimentally.

6. PLOS authors have the option to publish the peer review history of their article (what does this mean?). If published, this will include your full peer review and any attached files.

Reviewer #1: No

Reviewer #2: **Yes: **Dr. Rekha Vyas, Professor and Zonal Director Research, MPUAT, Udaipur

---

## [Author Response · Author response to Decision Letter 0]

9 Oct 2022

Response to Reviewer 1

1. Overall

1.1) The manuscript is very good and valuable, and the authors have done a lot of effort, but for me, the question is whether self-report and questionnaire without clinical examination accurately determine that the disease is caused by work?

Author to respond reviewer: Thank you very much for your kind understanding that this work was done a lot of effort, and we wish to contribute this valuable product to agricultural workers and related occupation around the world to be useful as soon as we can.

It was still limitation of self-report assessment on work-related MSDs. We identified the limitation at discussion section. However, the 4th question of MSFQ was confirmation of the work-related pain in the past month, from the previous study of the MSDs Severity and Frequency Questionnaire (MSFQ) (Chaiklieng 2019, PlosONE….[9] and was used by the previous screening of MSDs (available at https://www.mdpi.com/article/10.3390/safety8030061/s1, [14]. The previous studies used also this tool for self-report that could determine the occupational MSDs, and this study provide the available online tool that could be accessed easier and report result in such a real time. 

1.2) the following are suggested for improvement:

(1) There are some grammatical issues throughout the paper that would benefit from an additional review and in other words grammar needs to be corrected throughout the manuscript.

Author to respond reviewer: We are already proofreading before resubmission by native speaker.

(2) Introduction needs to be rewritten.

Author to respond reviewer: we revised with yellow highlight.

(3) The Materials and methods is long and vague, it is better to be concise and clear.

Author to respond reviewer: we revised with yellow highlight.

2. Abstract

2.1) The research method (Statistical population, Sampling method, etc.) should be written.

Author to respond reviewer: We already add with yellow highlight. However, abstract not exceed 300 words.

2.2) What tests have been used?

Author to respond reviewer: We already add with yellow highlight. However, abstract not exceed 300 words.

3. Introduction

3.1) The introduction and problem statement is very brief and requires further explanation.

Author to respond reviewer: we revised with yellow highlight in the first paragraph to be more explanation.

3.2) The importance of the topic should be explained in such a way that the designed tool can determine whether the disease is caused by the work or not.

Author to respond reviewer: we revised with yellow highlight which explain the MSFQ could determine the disease caused by work from the previous study and the screening phase in the second paragraph.

3.3) It would be better to include a literature review.

Author to respond reviewer: we revised with yellow highlight in the second paragraph to include the literature review.

3.4) Research innovation to be written

Author to respond reviewer: we revised with yellow highlight in the last paragraph of research innovation..

4. Materials and methods 

4.1) The research method should be expressed in a more coherent and clear way.

Author to respond reviewer: we revised with yellow highlight for more coherent method and clear way.

5. Discussions 

5.1) The discussion should be written to be reflect the results

Author to respond reviewer: we revised with yellow highlight in the part of discussion to be reflect the results.

6. Conclusions and suggestions 

6.1) The conclusion is very brief so it should be reinforced

Author to respond reviewer: we revised with yellow highlight more important points from the study

6.2) Research limitations, the strengths and weaknesses of the study, suggestions should be written

Author to respond reviewer: we revised with yellow highlight in the second paragraph for the limitation and the strength of the study.

7. Conclusions and suggestions 

7.1) The reference should be given at the end of the manuscript according to the journal format.

Author to respond reviewer: we followed by manuscript body formatting guidelines, modified April 2021.

7.2) Match the references inside the text and at the end of the text.

Author to respond reviewer: we rechecked all reference and format.

The Review Report (Reviewer 2)

Comments and Suggestions for Authors

The tool will help in self reporting of MSDs in agriculture and will help to cover workers all around the globe. It is always not possible to find musculoskeletal problems of farm workers experimentally.

Author to respond reviewer: Thank you for your kind comments and suggestions, this work was very hard working and needed a lot of effort and budget, and we hope that the fast publication will be very useful to move on the next step for contribution work to other agriculturist around around the globe

---

## [Decision Letter · Decision Letter 1]

31 Oct 2022

Self-reported musculoskeletal disorders questionnaire for agriculturists: An online self-assessment tool development

PONE-D-21-40986R1

Dear Dr. Sunisa Chaiklieng,

We’re pleased to inform you that your manuscript has been judged scientifically suitable for publication and will be formally accepted for publication once it meets all outstanding technical requirements.

Kind regards,

Mohammad Hayatun Nabi, MBBS, MHSM, MPH, PHD

Academic Editor

PLOS ONE

Additional Editor Comments (optional):

Reviewers' comments:

Reviewer's Responses to Questions

**Comments to the Author**

1. If the authors have adequately addressed your comments raised in a previous round of review and you feel that this manuscript is now acceptable for publication, you may indicate that here to bypass the “Comments to the Author” section, enter your conflict of interest statement in the “Confidential to Editor” section, and submit your "Accept" recommendation.

Reviewer #1: All comments have been addressed

Reviewer #2: All comments have been addressed

2. Is the manuscript technically sound, and do the data support the conclusions?

Reviewer #1: Yes

Reviewer #2: Yes

3. Has the statistical analysis been performed appropriately and rigorously? 

Reviewer #1: Yes

Reviewer #2: Yes

4. Have the authors made all data underlying the findings in their manuscript fully available?

Reviewer #1: Yes

Reviewer #2: Yes

5. Is the manuscript presented in an intelligible fashion and written in standard English?

Reviewer #1: Yes

Reviewer #2: Yes

6. Review Comments to the Author

Reviewer #1: (No Response)

Reviewer #2: Work is quite extensive and good and can be considered for further needful publication. This self reported questionnaire will be helpful to know MSDs without the use of any sophisticated equipment.

7. PLOS authors have the option to publish the peer review history of their article (what does this mean?). If published, this will include your full peer review and any attached files.

Reviewer #1: No

Reviewer #2: **Yes: **Dr. Rekha Vyas, Professor

---

## [Editor Report · Acceptance letter]

12 Dec 2022

PONE-D-21-40986R1 

Self-reported musculoskeletal disorders questionnaire for agriculturists: An online self-assessment tool development 

Dear Dr. Chaiklieng:

I'm pleased to inform you that your manuscript has been deemed suitable for publication in PLOS ONE. Congratulations! Your manuscript is now with our production department. 

Kind regards, 

on behalf of

Dr. Mohammad Hayatun Nabi 

Academic Editor

PLOS ONE